# **Sub-grid Parameterization of Iceberg Drag in a Coupled Iceberg-Ocean Model**

Paul T Summers<sup>1,2</sup>, Rebecca H Jackson<sup>2,3</sup>, and Alexander A Robel<sup>1</sup>

**Correspondence:** Paul T Summers (paul.summers@rutgers.edu)

#### Abstract.

Ocean conditions in fjords play a key role in the accelerating ice mass loss of Greenland's marine terminating glaciers. Ice mélange and icebergs have been shown to impact fjord circulation, heat and freshwater fluxes, and the submarine melting of glacier termini. Previous attempts to model icebergs largely fall into two camps: small-scale models that resolve icebergs and represent the impact of form drag, and larger-scale models that parameterize sub-grid-scale icebergs but neglect iceberg drag. Here, we develop an extension of the large-scale style *iceberg* package for the MIT general circulation model (MITgcm) to implement a novel, scalable parameterization to incorporate the impact of iceberg drag while also improving overall computational performance of the *iceberg* package by  $\sim 90\%$ . To demonstrate our parameterization, we benchmark our method against existing iceberg-resolving models and compare to the previous configuration of *iceberg*. With the inclusion of sub-grid-scale drag, our model skillfully reproduces ocean conditions and iceberg melt rates of iceberg-resolving models, while reducing computational cost by orders of magnitude. When applied to a multi-month fjord-scale simulation, we find icebergs and iceberg drag have a significant impact on fjord and glacier-adjacent conditions, including cooling fjord waters and increasing circulation. We note that these effects are more moderate in the case of icebergs with drag, suggesting that studies without iceberg drag may overestimate the net impact of icebergs on the fjord system.

# 15 1 Introduction

Ice mass loss from Greenland is currently accelerating (Otosaka et al., 2023), and, for marine terminating glaciers, this mass loss is significantly influenced by ocean and fjord conditions (Slater et al., 2020). However, significant uncertainty remains in accurately simulating ocean circulation within fjords and the precise mechanism of this influence (Morlighem et al., 2019; Slater et al., 2020; Hager et al., 2024). An important component of this system is the melt and drag from icebergs, which can modify fjord conditions through freshening, cooling, increased upwelling, and modification of currents, as observed in field settings (Enderlin et al., 2016; Moon et al., 2018; Abib et al., 2024) and modeled in computational studies (Davison et al., 2020, 2022; Kajanto et al., 2023). Through these effects, icebergs can modify the near-glacier ocean conditions and thereby modulate submarine melting of the terminus (Davison et al., 2022; Hager et al., 2024). Furthermore, ice mélange, a dense rigid

<sup>&</sup>lt;sup>1</sup>Department of Earth and Atmospheric Sciences, Georgia Institute of Technology, Atlanta, Georgia, USA

<sup>&</sup>lt;sup>2</sup>Department Marine and Coastal Sciences, Rutgers University, New Brunswick, New Jersey, USA

<sup>&</sup>lt;sup>3</sup>Department of Earth and Climate Sciences, Tufts University, Medford, Massachusetts, USA

pack of icebergs and sea ice, can play an important role providing buttressing stress to the glacier and can influence the calving rate (Amundson et al., 2010; Robel, 2017; Schlemm and Levermann, 2021; Amundson et al., 2025).

Icebergs and ice mélange play a particularly important role in Greenland, where many fjords are home to upwards of 10,000 icebergs at any time. In Sermilik and Ilulissat Fjords in Greenland, icebergs and ice mélange have been shown to contribute over  $1000\,\mathrm{m}^3/\mathrm{s}$  of freshwater flux, up to 50% of the total freshwater flux delivered to the fjord (Enderlin et al., 2016; Moyer et al., 2019). This freshwater flux from icebergs and ice mélange melt vastly outweighs contributions from terminus melt, and is comparable with subglacial discharge (Jackson and Straneo, 2016; Moon et al., 2018). Ice mélange is typically found within 10s of kilometers from the glacier terminus, but iceberg melt can contribute significant freshwater flux even 100+ km away from the glacier terminus (Moyer et al., 2019).

The melt rates of glacier fronts and icebergs are particularly sensitive to ocean velocities (Jenkins, 2011; FitzMaurice et al., 2017; Schild et al., 2021; Cenedese and Straneo, 2023; Zhao et al., 2024), and therefore it is important to realistically capture factors affecting fjord velocities. Iceberg-resolving models, which are high resolution models that can resolve individual icebergs with grids of  $\Delta x, \Delta y \sim 10$  meters, have shown that icebergs impact fjord circulation through a form drag effect that can reduce velocities within an iceberg mélange by over 90% (Hughes, 2022), even when the icebergs themselves have no skin drag (i.e. a free slip condition). This is important because the *iceberg* MITgcm package, a widely used parallelized numerical model for modeling iceberg thermodynamics and effects on ocean circulation, does not yet include the effect of such form drag. Thus, previous studies (Davison et al., 2020, 2022; Kajanto et al., 2023; Hager et al., 2024) have not included the impact of form drag from icebergs on ocean currents, which has been shown to be an important feedback process in the coupled iceberg-ocean system (Hughes, 2024).

Explicit representation of individual icebergs and their interaction with ocean circulation, as in Hughes (2024) and Jain et al. (2025), requires fine horizontal model resolution ( $10^1$  meters), and thus is computationally expensive to deploy in a fjord-scale model over climatically relevant timescales (months to centuries). Multi-year simulations of ice mélange have previously neglected side-melting and form drag of icebergs using the *shelfice* MITgcm package (Wood et al., 2025). Thus, a parameterization that can accurately represent iceberg-scale drag effects in coarser resolution fjord- and regional-scale models  $(10^2 - 10^3 \text{ meters})$  is essential to capture the influence of icebergs on ocean circulation and near-glacier properties, over seasonal to multi-decadal variations in ocean, atmosphere and glacier conditions. In this study, we develop and demonstrate a new extension of the *iceberg* package, which we refer to as *iceberg2*, that includes representation of sub-grid scale iceberg drag effects on ocean circulation. We compare our coarse resolution parameterization against iceberg resolving models that specifically target the mechanical blocking and drag effect, as well as a full thermodynamic case. Additionally, we apply this new drag-enabled *iceberg2* package to a multi-month fjord scale model to demonstrate the impact of icebergs and iceberg drag on fjord dynamics for one particular idealized scenario.

Figure 1. Simplified schematic of the *iceberg2* package functionality within the MITgcm.  $\Delta x, \Delta y, \Delta z$  are the resolution of the grid in MITgcm, and our schematic is drawn for a single  $\Delta x \times \Delta y$  slice of the water column. Individual iceberg geometries are stored for the modeling of melt processes (upper center) while grid-averaged values of ice volume fraction  $\varphi$  are used to calculate blocking and drag values at each depth layer (lower center). State variable tendencies are then passed to the primary MITgcm solver and conditions are evolved for the next time step.

## 55 2 Sub-grid Parameterization of Iceberg Drag Effect

To enable regional scale (horizontal grid spacing > 200 meters) modeling, we adopt a hybrid approach to our representation of individual icebergs in our model, building off the preexisting *iceberg* package (Davison et al., 2020). In *iceberg*, the discrete geometry of each sub-grid iceberg is fully resolved for thermodynamic modeling. The rectangular dimensions (draft, width, length) of discrete icebergs within each grid (X,Y location) are inputs to the model specified at initialization, allowing for an arbitrary number of icebergs per grid (X,Y location) (Figure 1). These geometries and locations are held constant in time, thus icebergs never move cells, nor change geometry from melting. At each time step, the melt rate is calculated using the three-equation melt parameterization (Jenkins, 2011) on each face and every depth level of every iceberg using the ambient ocean conditions within that MITgcm cell (X,Y,Z location). Following Cowton et al. (2015) and many others, a minimum melt velocity is imposed to parameterize the effect of ambient melt plumes. Freshwater flux, salt and heat tendencies are summed across all icebergs within the cell (X,Y,Z location), and then passed to the MITgcm solver at every timestep. Diagnostics of these time-varying values (freshwater flux, melt rate, and heat flux) can also be saved at each timestep. Additionally, the melt

parameterization is built such that it can account for the velocity of each iceberg drifting with the average ocean velocity along its entire draft, where  $u_{\text{drift}} = \sum_{i \leq k} (u_i d_i) / \sum_{i \leq k} (d_i)$ . Then the velocity used for melt at every depth is  $u_i^{\text{melt}} = u_i - u_{\text{drift}}$ , where  $u_i$  is the ocean velocity at layer i,  $d_i$  is the thickness of layer i, and the sum ranges over all  $i \leq k$  where k is the deepest layer the iceberg reaches. In this study we focus on icebergs fixed within a mélange and thus do not utilize this drifting option, but we keep it available for use in iceberg2.

In development of *iceberg2*, we do not adjust the thermodynamic components described above, but we do adjust the implementation for faster computational efficiency (Appendix A). This results in  $\sim 90\%$  faster computational performance in *iceberg2* when considering melt alone (no drag), but the equations for calculating melt are not changed. The per-iceberg approach above is valid for thermal and freshwater contributions as icebergs add linearly in heat and freshwater flux, but this linear behavior does not apply for drag (Hughes, 2022). In other words, the drag exerted by a pack of icebergs is not the linear sum of the drag from each individual iceberg, since the presence of each iceberg impacts the flow conditions around other nearby icebergs.

In order to add grid-scale mechanical coupling to *iceberg2*, the iceberg geometry is reduced to a volume fraction occupied by icebergs,  $\varphi$ , for each vertical layer in every cell:

$$\varphi = \frac{1}{\Delta x \Delta y \Delta z} \sum_{i} W_{i} \times L_{i} \times H_{i}$$
(1)

where the sum is over all icebergs i within a grid cell, with widths  $W_i$  and lengths  $L_i$ , and  $H_i$  is depth the iceberg extends into the cell centered at depth z (Figure 1).  $\Delta x$  and  $\Delta y$  are the horizontal grid spacing of the ocean model, and  $\Delta z$  is the vertical spacing.  $\varphi$  is used to calculate the parameterized bulk drag effect of all the icebergs in the cell.  $\varphi$  is very similar to the iceberg surface area fraction  $\lambda$  discussed in other studies (e.g. Hughes (2022, 2024))

$$\lambda = \frac{1}{\Delta x \Delta y} \sum_{i} W_{i} \times L_{i} \tag{2}$$

where now i sums over all icebergs at the surface.  $\lambda$  is only defined at the surface z=0. When all icebergs in a cell extend through the full thickness of the layer centered at depth z, for example if the surface layer is thinner than the shallowest draft,  $\varphi(z)=\lambda$ . For other depths,  $\varphi$  is calculated by carrying out the summation in equation 1 at all depths and for all icebergs extending into that depth layer. As observations of icebergs at the ocean surface are most readily available, we create our iceberg distributions based on a desired averaged iceberg surface area fraction,  $\overline{\lambda}$ , and then calculate  $\varphi$  using equation 1 and the geometry of our iceberg distribution, which is described in more detail in Section 2.1.

In our model approach, there are two processes by which icebergs physically interact with the ocean: physical blocking and bulk form drag. For iceberg-resolving models, these two effects are resolved and act together by not allowing ocean flow through iceberg cells (e.g. Hughes (2022)), but for our parameterization of sub-grid scale icebergs we must account for each distinct process. We include the effect of physical blocking by leveraging partially filled cells with the MITgcm grid (Adcroft et al., 1997), which we detail in Appendix A. Although the previous implementation of *iceberg* intended to include this blocking effect, all previous studies using *iceberg* inadvertently had the blocking effect disabled (Davison et al., 2020, 2022; Kajanto et al., 2023; Hager et al., 2024; Slater et al., 2025). The blocking effect tends to accelerate the ocean currents as they pass

through the reduced open volume of partially filled cells, and, without an additional drag parameterization, blocking-only results in non-physical acceleration of ocean currents passing through an ice mélange, which we discuss in Section 4.2.

To include the effect of form drag of many icebergs, we parameterize the bulk form drag of all icebergs within each cell at every depth layer using a form of the drag equation that can span the limits of a single iceberg to a channel wide blockage as discussed in Hughes (2022)

105 
$$\tau_{\mathbf{d}} = \rho C_{bd} \mathbf{u}^{\alpha(\varphi)} \Delta z \beta(\varphi)$$
 (3)

where  $\tau_{\mathbf{d}}$  is the net drag stress across the layer,  $\rho$  is the density of the ocean,  $C_{bd}$  is a bulk form drag parameter,  $\Delta z$  is the vertical grid spacing of the model or model layer, and  $\mathbf{u}$  is the ocean velocity. This drag formulation assumes icebergs are fixed in a static mélange and not freely drifting.  $\alpha(\varphi)$  is the parameterized power law scaling of velocity, and  $\beta(\varphi)$  is the parameterized filling fraction, capturing the effective frontal area of all the icebergs within the cell.  $\beta$  captures how effectively icebergs obstruct open flow pathways through the cell where  $\beta=0$  represents completely unobstructed flow and  $\beta=1$  means no unobstructed pathways exist. These functions  $\alpha,\beta$  can take many forms and we build 3 options for both  $\alpha(\varphi)$  and  $\beta(\varphi)$  into the *iceberg2* package motivated by Hughes (2022).

We build the scaling exponent for velocity in our drag parameterization with 3 regimes:

$$\alpha(\varphi) = \begin{cases} 1 & \text{Linear} \\ 2 & \text{Quadratic} \\ 1 + .75 * (1 - \varphi) & \text{Hybrid (default).} \end{cases}$$
 (4)

The case of drag stress varying linearly with velocity best describes a full-width blockage in a stratified fluid (Klymak et al., 2010), while a quadratic relationship is more typical for an isolated obstacle, as described in Hughes (2022). When considering a large pack of obstacles in a stratified fluid, the effective power law tends to decrease from  $1.75 \rightarrow 1$  as  $\varphi$  increases from  $0 \rightarrow 1$ , so we implement a "Hybrid" form of  $\alpha(\varphi)$  to capture the transition described in Hughes (2022) (Figure A1 a). We use this as our default form of  $\alpha$ . We do consider other forms of  $\alpha(\varphi)$  that capture the curvature of the transition as shown in Hughes (2022), like a cubic fit, but we find results to be insensitive to this level of fitting and so defer to the simpler linear fit. For the parameterized filling fraction we again include three regimes:

$$\beta(\varphi) = \begin{cases} \varphi & \text{Linear} \\ -(\varphi - 1)^2 + 1 & \text{Quadratic} \\ -(\varphi - 1)^4 + 1 & \text{Quartic (default)}. \end{cases}$$
 (5)

As the cell becomes more filled with icebergs, each additional iceberg increases  $\varphi$  but does not necessarily increase the frontal area that contributes to drag (e.g. an iceberg immediately in the lee of another iceberg adds little additional drag). We again build in 3 cases, where the linear  $\beta$  case assumes no shadowing effect and icebergs do not block each other (i.e., the limit in which icebergs fill perfectly in the cell in the direction transverse to flow). The quadratic case follows the observed form of the fit in Hughes (2022) allowing the cell to fill rapidly at the start and saturates to full at  $\varphi = 1$ . The case of quartic  $\beta$  similarly

rapidly fills at low  $\varphi$  but now saturates to > .95 by  $\varphi = 0.6$  (Figure A1 b), consistent with the observation of Hughes (2022) that after  $\varphi(z=0) = \lambda \approx 0.6$  the entire frontal area is typically blocked and additional filling does not increase the frontal area. We use this quartic case as our default form of  $\beta$ . We summarize the variables we have introduced in Table 1 and plot  $\alpha(\varphi), \beta(\varphi)$  in Appendix A.

We find the model is not uniquely sensitive to the exact choice of  $\alpha(\varphi), \beta(\varphi)$ , as tuning  $C_{bd}$  can significantly compensate for the particular choice of  $\alpha(\varphi), \beta(\varphi)$ . However, we find that our recommended choices of Hybrid  $\alpha(\varphi)$  and Quartic  $\beta(\varphi)$  result in acceptable parameterization performance (discussed in detail below) across the entire range of  $\overline{\lambda}, U$  we consider. We build in user control of  $\alpha, \beta$  to allow flexibility when users are intentionally using the model to explore specific regions of parameter space, but we caution that recalibration of  $C_{bd}$  is warranted for such use.

We note that our bulk form drag parameter  $C_{bd}$  has a similar form and value to the skin friction drag parameter used in melt parameterizations Jenkins (2011), but they should not be confused. The  $C_{bd}$  parameter here captures the independent process of bulk form drag of many icebergs and should not be confused with skin friction drag on a surface (e.g. Klymak et al. (2021)). Thus, discussions of adjustment to the  $C_d$  value from Jenkins (2011), like those in Zhao et al. (2024), do not apply to the bulk form drag parameter  $C_{bd}$  we discuss here.

# 2.1 Iceberg Geometries

We produce our iceberg distributions following a power-law distribution matching observations of icebergs (Enderlin et al., 2016; Sulak et al., 2017) in which the relative number of icebergs (N) depends on the horizontal area of the iceberg (A) and so varies according to:  $N \sim A^{-1.9}$ . We generate these icebergs by randomly generating iceberg areas from this power law distribution until we reach sufficiently close to the desired total horizontal area of icebergs (within 0.5%), and then randomly distribute those icebergs across iceberg containing cells within our domain. In this distribution process, some cells become slightly overfull (i.e. above the average iceberg surface area fraction  $\overline{\lambda}$ ) as this captures some of the variability of the randomly placed icebergs in iceberg resolving models, as well as the non-uniform distributions of naturally occurring icebergs.

We set the draft of every iceberg  $D_i$  as a function of its horizontal area  $A_i$ 

$$D_i = aA_i^{b-1} \tag{6}$$

where a, b are normally distributed variables with mean 6, 0.3 and standard deviation 1.22, 0.016 respectively (Sulak et al., 2017). Our results are particularly sensitive to statistics of the draft of our icebergs, so we encourage care in future studies in setting iceberg draft. In particular, setting an abrupt Heaviside-style minimum draft results in large numbers of icebergs that terminate in one particular depth cell and this leads to unrealistically high freshwater flux and vertical shear values for this layer. For this reason, we find it important to set draft as a randomly varying function of iceberg horizontal area. Due to the random distribution of our iceberg drafts, any stated minimum or maximum values are only restrictions on the mean of the distribution (equation 6), and the actual extreme values will likely extend beyond these values.

| Parameter         | Definition                                      | Notes                                              |  |
|-------------------|-------------------------------------------------|----------------------------------------------------|--|
| φ                 | Ice volume fraction                             | Calculated from iceberg geometry                   |  |
| λ                 | Ice area fraction at ocean surface              | Metric used to determine how 'full' a cell is      |  |
|                   |                                                 | with icebergs when placing icebergs in cells       |  |
| $\alpha(\varphi)$ | Scaling exponent for velocity in drag parame-   | Default Hybrid [Unitless]                          |  |
|                   | terization (Equation 3)                         |                                                    |  |
| $\beta(\varphi)$  | Parameterization of filling factor used in drag | Default Quartic [Unitless]                         |  |
|                   | parameterization (Equation 3)                   |                                                    |  |
| $C_{bd}$          | Bulk form drag coefficient in drag parameteri-  | Default 0.0025 [Units $m^{1-\alpha}s^{\alpha-2}$ ] |  |
|                   | zation (Equation 3)                             |                                                    |  |

Table 1. Summary of parameters definitions and defaults used in this study

# 3 Application to Ocean Modeling in the MITgcm

We benchmark the new drag and blocking components of the *iceberg2* package against Hughes (2022), an iceberg-resolving ocean model in the MITgcm that omits the effect of melt, across a range of average iceberg surface area fractions, λ, and forcing current speeds, U. This initial benchmarking is done with iceberg melt disabled, following Hughes (2022). We also use this same geometry to consider the impact of blocking alone (no drag) as well as the addition of iceberg melt. The implementation of these configurations is detailed in Appendix A. We then benchmark the full thermomechanical case of melting icebergs with and without iceberg drag against the thermomechanically coupled iceberg-resolving MITgcm study from Hughes (2024). Finally we run a fjord-scale domain over several months to demonstrate the functionality and scalability of our iceberg parameterization.

#### 3.1 Model Domains

We set up our Forced Flow Domain following the configuration described by Hughes (2022) for a 2.4 km × 2.4 km mélange pack floating in a rectangular channel with vertical walls that is 32 km long, 2.4 km wide, and 600 meter deep, (Figure 2 a) subject to a forcing current that varies vertically (Figure 2 b).

$$U(z) = U\cos\left(\frac{\pi z}{600}\right) \tag{7}$$

The fjord is initialized with a uniform salinity and linearly varying temperature (Figure 2 b), and with a linear equation of state. This creates a linear density profile such that the Brunt-Väisälä frequency,  $N = 5.2 \times 10^{-3} \text{ s}^{-1}$ , is comparable to average values for Greenlandic fjords (e.g. Sanchez et al. (2023)). In Hughes (2022), this density gradient is produced by a temperature gradient alone to allow for disabling of the salt solver in MITgcm, which increases computational efficiency. To produce a realistic density gradient, Hughes (2022) set the temperature field to nonphysical values, with temperatures as low

| Name               | Domain                      | Icebergs       | Blocking | Drag | Melt |
|--------------------|-----------------------------|----------------|----------|------|------|
| H22                | Hughes (2022)               | Fully Resolved | Yes      | Yes  | No   |
| FF_NoMelt          | Forced Flow (Based on H22)  | Parameterized  | Yes      | Yes  | No   |
| FF                 | Forced Flow (Based on H22)  | Parameterized  | Yes      | Yes  | Yes  |
| FF_NoDragMelt      | Forced Flow (Based on H22)  | Parameterized  | Yes      | No   | No   |
| FF_NoDrag          | Forced Flow (Based on H22)  | Parameterized  | Yes      | No   | Yes  |
| FF_NoBlockDragMelt | Forced Flow (Based on H22)  | Parameterized  | No       | No   | No   |
| FF_NoBlockDrag     | Forced Flow (Based on H22)  | Parameterized  | No       | No   | Yes  |
| H24                | Hughes (2024)               | Fully Resolved | Yes      | Yes  | Yes  |
| MM                 | Mélange Melt (Based on H24) | Parameterized  | Yes      | Yes  | Yes  |
| MM_NoBlockDrag     | Mélange Melt (Based on H24) | Parameterized  | No       | No   | Yes  |
| FJ_NoIcebergs      | Fjord Scale                 | None           | No       | No   | Yes  |
| FJ                 | Fjord Scale                 | Parameterized  | Yes      | Yes  | Yes  |
| FJ_NoBlockDrag     | Fjord Scale                 | Parameterized  | Yes      | Yes  | Yes  |

**Table 2.** Full list of cases, model domains, and physics resolved in each case discussed.



as -8.3°C for liquid water. We follow this configuration to match Hughes (2022) for this benchmarking exercise, however we note that the equivalent density gradient from a more physical salt gradient (0°C, 34 PSU at the surface - 36.24 PSU at 600 meters) was also tested and produces identical results for model runs with no iceberg melt. We use a coarser spatial resolution of  $\Delta x = \Delta y = 200$  meters (compared to 10 meters in Hughes (2022)) as well as a slightly coarser vertical resolution with 50 layers, Nr = 50, (compared to 64). As a result of this coarser resolution, CFL (Courant–Friedrichs–Lewy) constraints are weaker, and we can use a longer time step of 15-25 seconds (instead of 1-2 seconds). We enable implicit viscosity and diffusivity using a 3D Smagorinsky scheme (Smagorinsky, 1963), with background values of vertical and horizontal viscosity set to  $10^{-4}$ ,  $10^{-3}$  m<sup>2</sup>/s respectively, and background diffusivity of  $10^{-5}$  m<sup>2</sup>/s.

We spin up the forced flow model using the same 8 hr spin-up process as Hughes (2022): using a modified rbcs package, the first 4 hours the entire domain is subject to a restoring force, followed by a 4 hour linear ramp down in restoring force strength, and then for the remaining model run the restoring force is only present in 8 km sponge regions at the east and west boundaries of our domain. Sidewalls and the bottom are set to a free slip kinematic boundary condition. Given the reduced computational needs of our model, we run all simulations for 3 days (compared to 36 hours in Hughes (2022)) which we find further reduces transient effects in the results but does not significantly impact our findings.

We set a minimum iceberg width of 40 meters and maximum depth of  $\sim 140$  meters to match the distribution of icebergs in Hughes (2022). Although we use very similar statistics in our generation and placement of icebergs, our iceberg fields are not identical so some variability of our results may be explained by random difference in iceberg distributions. To model melt of icebergs and ice mélange, we follow the "High Melt" case of Hughes (2024) using the three equation melt parameterization (Jenkins, 2011) specifying  $\lambda_1 = -5.75 \times 10^{-2} \, ^{\circ}\text{C}$ ,  $\lambda_2 = 9.01 \times 10^{-2} \, ^{\circ}\text{C}$ ,  $\lambda_3 = 7.61 \times 10^{-4} \, ^{\circ}\text{C/m}$ ,  $\gamma_t = 4.4 \times 10^{-3}$ ,  $\gamma_s = 4.4 \times 10^{-3}$ ,  $\gamma_t = 4.4 \times$ 

**Figure 2.** Model domains and boundary conditions for our three model configuration. (a,b) Bathymetry and forcing conditions for the forced flow domain following Hughes (2022). (c,d) Bathymetry and initial/boundary conditions for the mélange melt domain following Hughes (2024). (e,f) Bathymetry and initial/boundary conditions for our fjord scale domain. In a,c,e gray shading indicates the location of the ice mélange and black regions are the wall boundaries of bathymetry = 0.

 $1.24 \times 10^{-4}$  and the minimum velocity of  $u_{\min} = 0.04 \,\mathrm{m/s}$ . We follow Hughes (2024) in our notation, using  $\gamma_{t,s}$  to directly represent the Stanton number (St), instead of using formulations involving  $\Gamma_{t,s}$ , but note that these are related terms with  $\gamma_{t,s} = \sqrt{C_d} \Gamma_{t,s} = St$ .




For our second benchmark, the Mélange Melt Domain, we follow the configuration described by Hughes (2024) for a 8 km  $\times$  5 km mélange pack of average surface ice fraction,  $\overline{\lambda}=0.10$ , in a 600 meter deep, vertically walled channel. The west end of the channel is abutted by a glacier terminus, where ambient melt is modeled by the *iceplume* package Cowton et al. (2015). We again use a coarser spacial resolution of  $\Delta x=\Delta y=200$  meters (compared to 10 meters in Hughes (2024)) as well as a slightly coarser vertical resolution Nr=50, compared to 64. The walls are vertical with a free slip kinematic boundary condition. We extend our domain 80 km to the east, longer than the 35 km in Hughes (2024) to reduce the effect of boundary conditions. We impose an along-coast current in the eastern 5 km of our domain where there is an open boundary implemented with the open boundary conditions for regional model package (*obcs*). The mélange floats in a uniform 2°C ocean with a linear vertical salinity gradient (Figure 2 c,d) and we now use a non-linear equation of state (Jackett and Mcdougall, 1995). Following Hughes (2024) icebergs maximum draft is set to  $\sim$  200 meters. We run this simulation for 7 days, with no other external forcing. Otherwise, model settings are the same as the Forced Flow Domain.

For our final Fjord Scale Domain, we use a 5 km wide, vertically walled 600 meter deep, 80 km long fjord with a glacier terminus on the western end. The mélange is 15 km long, 5 km wide, and has linearly decreasing  $\overline{\lambda} = 0.60 - 0.01$  from 0 to 15 km along fjord. We initialize the fjord (Figure 2 e,f) with linear temperature and salinity profiles which are forced at the 5 km

wide eastern open boundary where there is again an along-coast current. We allow for ambient melt across the glacier terminus and force the system with a subglacial discharge through a half-conical plume, initialized with  $500 \text{ m}^3/\text{s}$  of freshwater at the bottom of the glacier (z=-600 and x=0 meters) implemented with *iceplume* (Cowton et al., 2015). We use a coarser horizontal resolution than previous simulations of  $\Delta x = \Delta y = 400$  meters, and simulate 200 days of melt. Otherwise, model settings are the same as the Mélange Melt Domain.

#### 4 Forced Flow Domain Results

Figure 3. The configuration of our model domain for the  $\overline{\lambda}=0.20,\,U=0.12$  m/s case. (a) Distribution of  $\lambda$  across the mélange pack. (b) Depth variance of  $\varphi$ . Light gray shows  $\varphi(z)$  of every cell containing icebergs, and black dashed line shows the average  $\varphi$  over the entire mélange area,  $\overline{\varphi}(z)$ . (c) Histogram of iceberg drafts. (d) Width-averaged velocity of our model domain at the end of simulation. Cells are shaded with the width-averaged value of  $\varphi$  to illustrate the location of the mélange pack.

We test the correspondence between our parameterized model and the iceberg-resolving model of Hughes (2022) by exploring a range of iceberg surface area fraction  $\overline{\lambda}$  and forcing current velocity U. Hughes (2022) does not consider melt effects, so we implement FF\_NoMelt style model runs for comparison. We use the free parameter  $C_{bd}$  to tune these model runs, but select one constant value of  $C_{bd} = 0.0025$  across all runs shown here (our tuning process is described in Appendix B). We find that this value of  $C_{bd}$  results in a good fit across all  $\overline{\lambda}$ , U values considered. We note that for a given range of  $\overline{\lambda}$ , U we expect  $C_{bd}$  to depend on the choice of  $\alpha(\varphi)$ ,  $\beta(\varphi)$ , but we do not explore that region of parameter space here.

The primary metric of comparison between our parameterized model and the iceberg-resolving model is the mean modeled ocean velocity within the mélange pack, as this is the first-order control on iceberg melt rates. To calculate this velocity, we

average over the non-iceberg volume of each cell across the entire mélange pack, and extract the velocity above 300 meters depth as show in Figure 4. For each set of  $\overline{\lambda}$  and U values, we plot the mean velocity within the ice mélange, normalized by maximum driving velocity (i.e.  $\frac{u(z)}{U}$ ), for both our parameterized model (solid line) and the iceberg-resolving model of Hughes (2022) (dashed line). The normalized driving velocity,  $\frac{U(z)}{U}$ , is plotted as a black dotted line. We also plot the median and 90th percentile for iceberg draft in the  $\overline{\lambda}=0.2$  case with red dashed lines.




Our coarse-resolution model replicates the important features of the iceberg-resolving model. In both models, the presence of the melange causes a significant decrease in velocity near the surface, compared to the forcing velocity at this depth. While iceberg drag slows velocity near the surface, there is an enhancement of flow in both models below the deeper drafts (90th percentile) of the mélange. This large scale behavior is response to the ice mélange acting as a permeable channel wide blockage: forcing a portion of flow under the effective depth of the mélange, and another portion flowing through the ice mélange via tortuous paths at a reduced speed.

To quantify the difference between our model and Hughes iceberg-resolving model, we plot the relative residual

$$\epsilon(z) = \frac{u(z) - u^H(z)}{U(z)} \tag{8}$$

and show the root mean squared error (RMSE, over the upper 100 meters and upper 275 meters of the domain) in the legend of Figure 4 c, d

$$RMSE_{100} = \sqrt{\sum_{i=1}^{N_{100}} \frac{\epsilon(z_i)^2}{N_{100}}}, \quad RMSE_{275} = \sqrt{\sum_{i=1}^{N_{275}} \frac{\epsilon(z_i)^2}{N_{275}}}$$
(9)

where  $u^H(z)$  is the velocity from the iceberg-resolving study Hughes (2022) and  $z_i$  is the depth of layer i, U(z) is the driving velocity (Equation 7) and  $N_{100}, N_{275}$  is the number of layers above 100, 275 meters depth respectively. We focus our consideration on the upper 100 meters (RMSE<sub>100</sub>) as this is the region representing more than 90% of all iceberg drafts, and where the vast majority of melt would occur. We also report RMSE<sub>275</sub> for those more concerned with ocean currents generically. We omit residuals near z=-300 meters since the forcing velocity U(z) goes to zero at this depth and thus the denominator goes to zero in the calculation of relative residuals. Below z=-300 meters, the flow is virtually unaffected by the melange (Figure 3 d) and thus we focus our analysis, like Hughes (2022), on the upper 300 m results. Across the entire  $\overline{\lambda}, U$  parameter space we explore, we generally find a good correspondence to Hughes (2022). Specifically, we find that velocity residuals within the upper 50 meters of the water column (median iceberg draft) are always less than 15% of driving velocity compared to Hughes (2022). Total root mean squared error for our model is 6% of driving velocity for the upper 100 meters of ocean (RMSE<sub>100</sub>) and 9% of driving velocity for the upper 275 meters of ocean (RMSE<sub>275</sub>, Appendix B).

The sinusoidal profile of velocity as a function of depth is expected for blocked flow in a stratified medium (Klymak et al., 2010), and the sinusoidal nature of our residuals arises from a mismatch in the wavelength of this effect in our model. Above the median iceberg draft, there is a region of positive residual velocities (faster velocities) across most runs at  $\sim 25$  meters depth. In contrast, within 15 meters the surface residuals are often negative ( $\epsilon \sim -0.05$ ) though this trend is not true at the bounds

of our parameter sweep (high  $\lambda$ , high and low U). Below the median draft of the mélange, residuals are generally negative (slower velocities) until below the deeper drafts of  $\sim 100$  meters. Again this trend doesn't hold for the most extreme values of our parameter sweep. The RMSE<sub>275</sub> values are generally higher than RMSE<sub>100</sub> as the magnitude of the velocity maximum underneath the mélange (100-150 meters in Figure 4 a, b) is generally not as well resolved by our parameterization compared to shallower waters (relative residual maximums in c, d between 100-150 meters).

Figure 4. Along-fjord ocean velocity averaged over the mélange pack for a range of  $\overline{\lambda}$  in a and range of U in b. Solid lines are this study, dashed lines are Hughes (2022) for the same conditions. Black dotted line is the driving velocity. Velocities are normalized by the driving velocity (U). Red dashed lines are the median and 90th percentile of iceberg draft for  $\lambda = 0.2$ . c,d show the relative residual  $\epsilon$  (equation 8) for a, b respectively, and list the RMSE<sub>100</sub>, RMSE<sub>275</sub> for each case in the legend. c, d share the color scale of a, b respectively.

#### 4.1 Sensitivity to Model Resolution


The utility of the parameterization we develop here is to capture the effects of a sub-grid scale process in a way that is computationally efficient for use in larger-scale modeling. Therefore, understanding how our parameterization error varies with grid resolution is crucial for its careful application to larger-scale models. To investigate this effect, we repeat the  $U = 0.12, \overline{\lambda} = 0.20$  case for a range of grid resolutions. We vary  $\Delta x$  from 100 to 2400 meters while fixing Nr = 50, and vary Nr = 0.20 case for a range of grid resolutions.

from 10 to 120 while fixing  $\Delta x = 200$  meters (Figure 5 A, B). While we change the grid for these runs, we do not change our tuning parameter  $C_{bd}$  or any other aspects of the MITgcm model configuration. We report the RMSE<sub>100</sub>, RMSE<sub>275</sub> for each case in the legend of panels c, d.

Figure 5. Ocean velocity averaged over the mélange pack for a range of Nr,  $\Delta x$  values for  $U=0.12, \overline{\lambda}=0.2$ . Black dotted line is the driving velocity, and the coarse gray line is the results from Hughes (2022) for the same  $U, \lambda$ . Red dashed lines are the median and 90th percentile of iceberg draft. c, d show the relative residual  $\epsilon$  (equation 8) for a, b respectively, and list the RMSE<sub>100</sub>, RMSE<sub>275</sub> for each case in the legend. c, d share the color scale of a, b respectively.

We find that the overall behavior of our parameterization is not significantly dependent on model resolution. We see the greatest reduction in RMSE<sub>100</sub> as horizontal grid resolution approaches the length of the largest icebergs (200 meters here) and when vertical resolution allows for more than 10 layers to resolve the full range of iceberg drafts (Nr = 25 here). Model performance depends on the number of vertical layers available to capture the sinusoidal variation of velocity with depth. The wavelength of this sinusoidal variation falls beneath the resolution of the coarsest vertical grids we consider here (Nr = 10, 12), thus RMSE<sub>275</sub> is particularly high (> 20% of driving velocity) for the coarsest Nr as the model is unable to resolve the deeper velocity maximum at  $\sim 200$  meters depth (Figure 5 b). For horizontal grid scales, as the grid size approaches the size of the largest icebergs our parameterization begins to have increasingly full cells, closely resembling the fully dry cells of the

iceberg-resolving model. In this limit, our parameterization begins to individually resolve the largest icebergs within the pack, and thus our model more closely matches the iceberg-resolving model. This illustrates how our parameterization is converging towards a model that fully resolves icebergs, and thus the residual decreases with increasingly fine horizontal resolution. However, once the horizontal grid size is more than double the largest icebergs ( $\Delta x = 400$  meters here), further coarsening does not significantly reduce the performance of our parameterization with a RMSE<sub>100</sub> of just 14% of driving velocity for the  $\Delta x = 2400$  meters case.

In summary, we find our parameterization of iceberg drag performs with satisfactory performance across a range of conditions and grid sizes. However, this benchmarking neglects the impact of freshwater production from melting, and is not yet clearly compared to existing drag-free iceberg models.

# 4.2 Impact of Blocking, Drag, and Melting

300

To investigate the impact of iceberg melting, we step outside the parameter space considered by Hughes (2022) and consider the impact of iceberg melting, drag, and blocking individually in this same idealized forced flow domain. By isolating each of these effects, we showcase the impact of each mechanism, and this also lets us compare our drag-enabled iceberg models against existing drag-free models (e.g. Davison et al. (2020)). Specifically, we consider  $\overline{\lambda} = 0.2$ , U = 0.12 m/s cases of FF\_NoBlockDragMelt, FF\_NoDragMelt, and FF\_NoMelt (Table 2). We consider each of these 3 cases without melting, as well as the corresponding 3 melt-enabled cases, FF\_NoBlockDrag, FF\_NoDrag, and FF. We plot the mélange averaged ocean velocities and net freshwater flux in Figure 6. The FF\_NoMelt case is the exact same as those in section 4, and the FF\_NoBlockDrag case is equivalent to that considered by Davison et al. (2020, 2022); Kajanto et al. (2023); Hager et al. (2024); Slater et al. (2025).

In the no melt cases, the competing effects of blocking and drag are well highlighted. The FF\_NoBlockDragMelt case reproduces the driving current very well as expected, as there is no iceberg interaction with the ocean for this case. The FF\_NoDragMelt case however accelerates up to 150% the driving velocity in the upper 100 meters, a result of the flow being squeezed by the reduced effective volume of iceberg filled cells, without any slowing effects of drag. In contrast, the FF\_NoMelt case slows flow in the upper 100 meters as discussed in Section 4 when both blocking and drag are applied. The oscillatory form of velocity as a function of depth for the FF\_NoDragMelt case follows a similar wavelength as seen in Section 4, but has the opposite sign with faster than driving velocity flow above 100 meters, and slower than driving velocity for 100-150 meters (Figure 6 a).

When we consider the effect of melt on these three cases, the FF\_NoBlockDrag case on average tracks the driving velocity, but exhibits a vertical oscillatory behavior again with wavelength comparable, but sign opposite to results in Section 4. The surface acceleration of the FF\_NoBlockDrag case is driven by the injection of meltwater within the mélange. The FF\_NoDrag case similarly sees an acceleration at the surface and slowdown from 10-50 meters depth compared to the FF\_NoDragMelt. Below 75 meters, the 90th percentile for iceberg draft, the FF\_NoBlockDrag and FF\_NoDrag cases become very similar in ocean velocity. The FF and FF\_NoMelt cases are very similar at all depths, as any impact of freshwater injection is immediately overcome by drag within the mélange.

Figure 6. (a) Ocean velocity averaged over the mélange pack for a range of blocking, drag, and melt options for U=0.12,  $\overline{\lambda}=0.2$ . Black dotted line is the driving velocity. Red dashed lines are the median and 90th percentile of iceberg draft. (b) Freshwater production rates, averaged over the mélange pack, per unit depth for a range of block and drag options. No melt cases are omitted. Red dashed lines are the median and 90th percentile of iceberg draft, black dashed line is the deepest draft, and the blue dashed line is the depth where the ambient temperature drops below the freezing point.

Considering the impact of blocking and drag on melt rate in Figure 6 b, we find that the freshwater flux varies by up to 250% between these cases. The greatest difference in freshwater flux is at the surface, where velocities are most different as well. As suggested by the velocity dependent form of our melt parameterization (Jenkins, 2011), the fastest case (FF\_NoDrag) produced the highest freshwater flux up to  $0.5\,\mathrm{m}^2/\mathrm{s}$  at the surface, while the FF case has the lowest freshwater flux of  $0.2\,\mathrm{m}^2/\mathrm{s}$  at the surface. The FF\_NoDrag and FF\_NoBlockDrag cases have very similar freshwater fluxes below 50 meters, the median iceberg draft. In all cases, freshwater flux decreases as the total surface area of icebergs diminishes with increasing depth. Additionally, there is reduced thermal forcing with increasing depth as the ambient temperature approaches the freezing point at z=-137 meters, the blue dashed line in Figure 6 b.

This section illustrates the impact of parameterized iceberg drag when modeling iceberg melt in an idealized scenario. Importantly, we find that for the FF case, results are very similar to the benchmarked FF\_NoMelt case. This gives us confidence that our drag parameterization can successfully be applied in conditions including iceberg melt, which we explore in the next section.

# 5 Mélange Melt Domain Results

325

330

We next consider the impact of iceberg drag on the coupled thermomechanical system of iceberg melting in a more realistic ocean domain. We again use results from an iceberg-resolving model model as a benchmark for comparison (Hughes, 2024), and run our coarser-scale model in a comparable geometry to evaluate its performance. For these cases with melt, we run our coarse model with melting enabled in two settings: with blocking and drag enabled (MM) and with blocking and drag

disabled (MM\_NoBlockDrag, setting barrierMask = False). This MM\_NoBlockDrag configuration is equivalent to the FF\_NoBlockDrag case above and previous studies using the *iceberg* package (e.g. Davison et al. (2020)). Details of this configuration are listed in Section 3.1 and visualized in Figure 7. Notably, this system is driven only by ambient iceberg melt and is not forced by any subglacial discharge or other prescribed ocean velocity.

Figure 7. The configuration of our model domain for the mélange melt with blocking case. (a) Distribution of  $\lambda$  across the mélange pack. (b) Depth variance of  $\varphi$ . Light gray shows  $\varphi(z)$  of every cell containing icebergs, and black dashed line shows the average  $\varphi$  over the entire mélange area,  $\overline{\varphi}(z)$ . (c) Histogram of iceberg drafts. (d) Width-averaged salinity of our model domain at the end of simulation. Cells are shaded with the width-averaged value of  $\varphi$  to illustrate the location of the mélange pack.

Figure 8. Comparison of mélange melt model results for z = -10 meters. Temperature (left) and along-fjord velocity (right) are plotted for three cases. Icebergs are plotted as gray in the iceberg resolving model of Hughes 2024 (a,b).

We compare our results directly against the results published by Hughes (2024), considering a horizontal slice at z=-10 meters depth as well as statistical results of ocean velocity and freshwater flux from melt. Figures 8 show temperature and along-fjord velocities for all 3 cases (H24, MM, and MM\_NoBlockDrag) for the horizontal slice z=-10 meters depth. Our full iceberg parameterization that includes drag and blocking (MM) replicates the main features of the H24, whereas the parameterization without blocking and drag (MM\_NoBlockDrag) shows significantly different signals in the circulation patterns and magnitude of flow. This is particularly evident in the southern region of our domain (y 

**Figure 9.** Impact of iceberg drag on meltwater production. (a) Total meltwater production over time, (b) Meltwater production over depth, (c) Histogram of ocean speed within the mélange pack, as well as whisker plots showing 10,25,50,75,90 percentiles. (d) Average ocean speed across the mélange pack as a function of depth. The red dash-dot line in c,d is the minimum melting velocity used in our melt parameterization. Panels b-d show values averaged over the final 2 hours of simulation.

the MM case, which both show a depth of maximum freshwater flux at 50 meters and a near-linear reduction in freshwater flux in the upper 50 meters (Figure 9 b). The MM\_NoBlockDrag case shows higher fresh water flux across the entire fjord depth, as well as a second local maximum of freshwater flux near the surface.



We report the ocean speed  $((u^2 + v^2 + w^2)^{1/2})$  across the ice mélange for the final 2 hours of simulation in a histogram of ocean speed for all ocean cells within the ice mélange (Figure 9 c) and as box and whisker plots showing 10,25,50,75,90 percentiles. We consider the depth variation of average ocean speed for the final 2 hours of simulation within the ice mélange in Figure 9 d as ocean speed is a primary control on melt rates (Jenkins, 2011). The red dash-dot line in Figure 9 c,d highlights the minimum melting velocity used in our melt parameterization, so variations in ocean speed above this values will drive changes in the parameterized melt rate. In Figure 9 c,d we report the H24 values for the entire mélange (gray) as well as only the cells that are in direct contact with ice and thus impact melt (dark gray). For H24 this is approximately 5% of the cells within the ice mélange. In our coarse model, all cells in the mélange pack are in contact with ice. The MM NoBlockDrag

case shows significantly faster ocean speeds compared to H24 melt-only speeds. The MM ocean speeds are very similar to the H24 melt-only speeds, but is slightly slower by 0.005 m/s for the 90 percentile value. In Figure 9 d we show that this slower ocean speed for the MM case compared to H24 melt-only speed is concentrated below 50 meters depth. Though all cases show a minimum in average ocean speed at around 30 meters, the MM\_NoBlockDrag case has consistently higher ocean speeds compared to H24 above 150 meters depth.

Thus, we have shown that our iceberg parameterization with full drag and blocking effects (MM) captures much of the behavior of the iceberg-solving model of H24, while at significantly coarser resolution. Further, the MM case significantly out performs the case with melt but no blocking/drag (MM NoBlockDrag).

Figure 10. The configuration of our model domain for the full fjord run. (a) Distribution of  $\lambda$  across the mélange pack. (b) Depth variance of  $\varphi$ . Light gray shows  $\varphi(z)$  of every cell containing icebergs, and black dashed line shows the average  $\varphi$  over the entire mélange area,  $\overline{\varphi}(z)$ . (c) Histogram of iceberg drafts. (d) Width-averaged temperature of our model domain at the end of simulation. Cells are shaded with the width-averaged value of  $\varphi$  to illustrate the location of the mélange pack.

# 6 Fjord Scale Domain Results


As a final case, we demonstrate the computational scalability of our parameterization for a quasi-realistic mélange-filled fjord system (Figure 10) driven by 500 m<sup>3</sup>/s of subglacial discharge (SGD). Details of this configuration are listed in Section 3.1. To compare the net effect of icebergs and iceberg drag, we now consider three cases: FJ\_NoIcebergs, were there is no melting, no blocking, and no drag from icebergs (i.e., no impact from icebergs and *iceberg2* package is disabled), FJ\_NoBlockDrag where we include iceberg melt with no blocking and no drag, and FJ where iceberg melt, blocking, and drag are all included. It is

important to note that almost all regional ocean model simulations in which Greenlandic fjords are resolved (Gladish et al., 2015; Carroll et al., 2017; Wood et al., 2024) are equivalent to the FJ\_NoIcebergs case. Relatedly, all prior studies using the *iceberg* package (Davison et al., 2020, 2022; Kajanto et al., 2023; Hager et al., 2024; Slater et al., 2025) are equivalent to the FJ\_NoBlockDrag case. We focus on three metrics: total freshwater flux, near-glacier temperature, and mid-fjord average velocity (Figure 11). Freshwater flux is the sum of subglacial discharge, glacier frontal melting, and iceberg/mélange melting; near-glacier temperature is the average temperature over the 800 meters (2 grid cells) of ocean closest to the glacier terminus (similar to Davison et al. (2022); Hager et al. (2024)); and the mid-fjord conditions are the along-fjord velocities spatially averaged over the region from 20 km to 40 km from the glacier. Near-glacier and mid-fjord and conditions are temporal averages of the final 18 days of our 200 day simulation. To highlight north/south asymmetry of flow in the mid-fjord region, we divide mid-fjord conditions into north and south halves of the fjord in Figure 11, as well as plot slices of the along fjord velocity, u, in Figure 12. We plot the map-view of u at z = -100 meters of the entire fjord (Figure 12 a,c,e), as well as the across fjord profile of the average u for the mid-fjord region in (Figure 12 b,d,f) for the FJ\_NoIcebergs, FJ\_NoBlockDrag, and FJ cases respectively.

Figure 11. Impact of icebergs and drag on fjord level values. (a) Total freshwater flux, including subglacial discharge as well as iceberg/mélange melt for the full 200 days. (b) Average temperature within 800 meters of the glacier terminus. (c) Average along-fjord ocean velocity in the mid-fjord region. Dashed lines show the northern half of the fjord, dotted lines show the southern half of the fjord, solid lines are the average across the entire width. (d) Average salinity within 800 meters of the glacier terminus averaged over the final 18 days. (e) Average ocean speed within 800 meters of the glacier terminus. (f) Glacier front melt rates, averaged across fjord, depth averaged values are also plotted in a dashed line. b-f are all averaged over the final 18 days. All frames share the color legend of panel b.

Figure 12. Comparison of Fjord Scale Domain results in map-view for z = -100 meters (a,c,e) and cross-sectional view of the midfjord region (b,d,f) for along-fjord velocity u. a,b show along-fjord velocity for the FJ\_NoIcebergs case. c,d show along-fjord velocity for the FJ\_NoBlockDrag case. e,f show along-fjord velocity for the FJ case. Gray shaded region in a,c,e is the midford region that is shown in panels b,d,f respectively. Gray dashed line in b,d,f is the -100 m depth that is shown in map-view in panels a,c,e respectively. In all panels the u = 0 contour is plotted as a thin gray line. All panels are temporally averaged over the final 18 days of simulation.

Our results show that icebergs significantly modify fiord conditions by increasing freshwater flux into the ocean, cooling glacier-adjacent conditions and modifying the overturning flow, in agreement with other studies of iceberg impacts on fjord dynamics (Enderlin et al., 2016; Davison et al., 2020, 2022; Kajanto et al., 2023; Hager et al., 2024). However, we note that the inclusion of iceberg drag impacts the magnitude of these effects. Namely, the FJ case has 0.25°C warmer near-glacier temperatures at 50 - 150 meters depth compared to the FJ NoBlockDrag case (Figure 11 b). This is the terminal height of the subglacial plume (velocity maximum in Figure 11 e), thus we argue this warmth arises from the plume becoming trapped within the mélange. The FJ case has slower and deeper return flow compared to the FJ NoBlockDrag case at 100-200 meters depth (Figure 11 c). The impact on mid-fjord flow is most apparent in comparing the average along-flow velocity of the northern and southern halves of the fjord, where we see a much smaller north/south velocity contrast, as well as overall slower flow in the FJ case compared to the FJ\_NoBlockDrag case (Figure 12). The finding of slower ocean velocities in the FJ case extends to the glacier front: Figure 11 e shows near-glacier ocean speeds with the FJ case having overall slower speeds compared to the FJ NoBlockDrag case. Figure 11 d shows the impact of icebergs on freshening near glacier water, but this trend is not significantly impacted by iceberg drag. The competing effects of warmer water and slower flow on glacier front melt rates largely cancel out for our study, with very slightly lower vertically averaged glacial melt rates for the FJ case compared to the FJ\_NoBlockDrag case, though both cases exhibit lower glacier melt rates that the FJ\_NoIcebergs case (Figure 11 f). For terminus melt parameterizations that are sensitive to ocean velocities below 0.04 m/s we would expect a much larger difference in glacial melt rates between the FJ and FJ NoBlockDrag cases. The full effect of mélange, ice drag, and subglacial discharge plumes on terminus melt rates is the subject of ongoing work and beyond the scope of this study.



Relatedly, the overall freshwater flux from iceberg melt is similar after 20+ days in the FJ\_NoBlockDrag and FJ cases (Figure 11 a). The initial disparity between these two (before 20 days) matches the disparity observed in Section 5, which we explained by much higher ocean velocities in the no drag case. After 20 days, the FJ\_NoBlockDrag cases still exhibits faster flow within the mélange, but this is balanced by colder and fresher conditions within the mélange (Figure 11 b), an effect that takes a few days to spin up, and thus was missed in our 7 day scenario. This match in freshwater flux only exists for the total net flux as the FJ\_NoBlockDrag case produces a large north/south asymmetry in freshwater flux across the mélange not present in the FJ case, as discussed in Appendix C.

Each fjord model was run on 20 cores across 2 nodes with a total wall time of 03:12:04 for the FJ\_NoIcebergs case, 04:28:52 for the FJ\_NoBlockDrag case (28% slower), and 04:28:26 for the FJ case (31% slower than FJ\_NoIcebergs, comparable to FJ\_NoBlockDrag). The FJ case should be more computationally expensive compared to the FJ\_NoBlockDrag, but this effect is less than the magnitude of random variability in model run times. Given the  $\sim 90\%$  speed-up we built into *iceberg2* and the minimal computational cost for including drag, *iceberg2* enables more performant fjord-scale simulations. To simulate our fjord with iceberg melt and parameterized iceberg drag (FF case), this equates to 13.3 core-hours per month of fjord simulation time, enabling scalable simulations at timescales of multiple months to multiple decades. We did not attempt to run an iceberg-resolving model like Hughes (2024) for this fjord-scale geometry, but scaling our FJ\_NoIcebergs case runtime linearly with number of grid cells and time steps for a  $\Delta x = \Delta y = 10$  meters, Nr = 64 run would result in a computational cost of over 17 core-years per month of fjord simulation time, roughly  $10,000\times$  the computational expense.

To summarize, we find that icebergs drive cooler, fresher, faster fjord conditions compared to an iceberg-free fjord, and iceberg drag moderates some of the effects. Namely, iceberg drag slightly warms and slows fjord conditions compared to models that lack iceberg drag.

## 430 7 Discussion





Icebergs and ice mélange have previously been modeled at a range of scales and realism, with widely varied computational demands. In this work, we develop a scalable parameterization to include the processes of melt as well as blocking and drag. Our model builds upon previous versions of *iceberg* (e.g. Davison et al. (2020)) and complements existing approaches of modeling icebergs and ice mélange, including iceberg-resolving models (Hughes, 2022, 2024; Jain et al., 2025), as well as simplified iceshelf-like approaches (Wood et al., 2025). Iceberg-resolving models are specifically designed for capturing the nature of ocean flow around icebergs, but due to computational requirements are not realistically scalable to fjord-scale, long duration simulations. Thus, we benchmark our model's performance against iceberg-resolving models and demonstrate that iceberg drag impacts important near-terminus and fjord-wide processes. Previous versions of *iceberg* have captured the 3D geometry of icebergs when resolving melt processes, but have omitted drag processes. We extend this package to include a parameterization of iceberg drag, and demonstrate the impact drag has on model results for an idealized fjord. Iceshelf-like approaches to modeling ice mélange utilize the *shelfice* MITgcm package which only resolves drag and melting at the bottom layer of the mélange, and completely blocks ocean flow within the mélange. This trade-off in realism comes with significant

computational efficiencies compared to iceberg-resolving models and previous versions of *iceberg*. Our work improves the computational efficiency of *iceberg2*, now offering computational performance of the same order of magnitude as iceshelf-like approaches. In limited testing (not shown), our model is  $\sim 150\%$  slower than iceshelf-like mélange modeling, while offering significantly improved model realism.






Iceberg and terminus melt rates are expected to be strongly dependent on the ocean velocity adjacent to the ice. These velocities are set by a combination of far-field (10-100s meters) ocean dynamics and also near-ice ambient melt plume speeds (Slater et al., 2015; FitzMaurice et al., 2017; Zhao et al., 2024). The effect of sub-grid scale ambient plumes is commonly included in models by imposing a minimum velocity to be used in the 3 equation melt parameterization (Slater et al., 2015; Zhao et al., 2024). For the simulations that we have run here, the velocities within the melange are relatively weak (mean  $\sim 0.02$  m/s) and often below this minimum velocity threshold. Thus, the melt rates we report are sensitive to the choice of this minimum velocity threshold, similar to Hughes (2024). However, the statistics of the ocean conditions (velocity and temperature) in the ice-adjacent cells is very similar between the icebergs-resolved simulations ( $\Delta x = 10$  meters) and our iceberg-parameterized runs ( $\Delta x = 200$  meters). The consistency of modeled iceberg-adjacent ocean conditions across modeling scales suggests that both  $\sim 10$  and  $\geq 10^2$  meter models (e.g. Hughes (2024); Jain et al. (2025), this study) produce reasonable far-field ocean conditions, but are still strongly dependent on their ambient plume parameterization for calculating melt rates.

A clear compromise of our coarser model, compared to iceberg-resolving models, is that it does not capture fine details of ocean flow. This is particularly evident in the lack of so-called "hot spots", discussed in Hughes (2024), regions with substantially faster flow (u > 0.05 m/s) where water squeezes between icebergs (Figure 8 a). However, Figure 9 c highlights how these "hot spots" of faster flow are disproportionally not in contact with icebergs, and the statistics of speed of ocean cells in contact with icebergs, which are actually used to calculate melt rates, (H24 (Melt Only)), are well reproduced by our coarse model MM (Figure 9 c,d). Overall, the MM case slightly under-predicts ocean speeds and melt rates (2.1% meltwater flux error), but it seems that the impact of these "hot spots" on melt rate may be moderated by processes providing physical insulation of icebergs from these fastest flows. An important finding here is that, though our melt parametrization enforces a minimum melt speed of 0.04 m/s, our MM case well matches ocean speeds in cells in contact with icebergs in the H24 case even below this speed (Figure 9 c). Thus, we expect our MM case to retain its skill at matching melt rates to the iceberg-resolving case H24 even subject to changes in this minimum melting speed, like those discussed in Zhao et al. (2024).

When we apply this model to a multi-month fjord run, we identify the net effects of iceberg melt, blocking, and drag on overall fjord conditions. Specifically, our results agree with previous studies that icebergs significantly increase overall freshwater flux (Enderlin et al., 2016), cool the near-glacier conditions (Davison et al., 2022; Hager et al., 2024), and increase the net exchange flux (Davison et al., 2020; Kajanto et al., 2023). Our results here show that all of these findings are modified by including iceberg blocking and drag (FJ), generally reducing the magnitude of each effect compared to numerical models without iceberg blocking and drag (FJ\_NoBlockDrag).

Without blocking and drag, we find that a strong cross-fjord asymmetry develops near-glacier conditions, with up to a 0.5°C temperature difference and a 60% difference in freshwater flux across fjord (Figure C1). However, cross-fjord gradients are substantially reduced when we include the most realistic effects of blocking and drag (FJ case) where there is almost no cross-

fjord variation in temperature or fresh water flux. These errors (faster flow, lower temperatures in FJ\_NoBlockDrag compared to FJ) could counteract each other in the calculation of melt, perhaps explaining why studies using the previous version of *iceberg* (Davison et al., 2022; Hager et al., 2024) yielded reasonable overall freshwater flux values, despite omitting iceberg drag. Though these errors roughly off-set for the FJ\_NoBlockDrag case here, is is not clear this is a reliable trend, and we show that drag-free simulations do not reliably reproduce the spatial distribution of mélange melt. Further, the distribution of mélange melt directly impacts mélange thickness, which could have significant effects on the mechanical strength of the mélange and thus impact glacier calving rates (Amundson et al., 2010; Robel, 2017; Amundson et al., 2025). Strong asymmetries in mélange melting would likely mechanically weaken the mélange more than uniform melting, particularly if melt is concentrated in bands near walls, which can contribute buttressing shear stresses (Robel, 2017; Amundson et al., 2025). Thus, we argue that iceberg drag is an important factor to consider for mélange melt dynamics, even if the spatially averaged melt rate is not significantly different between the specific cases, FJ and FJ NoBlockDrag, we consider here.

# 8 Conclusions




In this study, we describe a parameterization of the effect of sub-grid scale iceberg blocking and drag in the context of a 490 rigid ice mélange. To demonstrate the accuracy and computational performance of this parameterization, we implement it as an improvement to the *iceberg2* package in MITgcm and benchmark it against an iceberg-resolving model in idealized simulations. Our parametrization offers reasonable accuracy (RMSE<sub>100</sub> of 6% of driving velocity) across a wide range of parameter values and grid scales (Figures 4, 5) at drastically lower  $(1/10,000\times)$  computational expense. In our benchmarking case that includes iceberg melt, our parameterization of drag reproduces fjord-scale features with improved fidelity to the iceberg-resolving model 495 compared to a simulation without drag. Specifically, our parameterization of iceberg drag better matches freshwater production rate and overall velocity structure of the ocean compared to a simulation without drag. When we apply our model to a multimonth fjord-scale simulation, we find that icebergs cool, freshen and increase overall circulation with the fjord, in line with previous work (e.g. Davison et al. (2020); Kajanto et al. (2023); Hager et al. (2024)), though all of these effects are modified by 500 the inclusion of iceberg drag. Namely, iceberg drag suppresses ocean currents and slightly warms near glacier water compared to the no drag case when a subglacial plume is present. Overall, the net effect of icebergs and drag on fjord conditions are most apparent within the ice mélange, cooling near-surface waters and injecting freshwater, but effects are also seen in altered ocean currents > 10s km down fjord.

This work takes an important step toward more realistically capturing the complexities of coupled iceberg-ocean interactions within fjords, and enables more readily scalable computational methods for including iceberg drag effects at fjord and larger scales. We demonstrate the scalability of our method to a multi-month fjord scale model run, and the importance of including the effect of both iceberg melt and iceberg drag on overall fjord dynamics. This innovation enables future efforts to simulate the co-evolution of icebergs and ocean circulation near ice sheets on multi-decadal and longer time scales.

Code and data availability. The iceberg drag enabled version of *iceberg2* is available at https://zenodo.org/records/14721713 (Summers, 2025). Example model runs, as well as all data and scripts to reproduce all figures can be found at https://zenodo.org/records/15116445 (Summers et al., 2025).

# Appendix A: Computational Details and Improvements




We include the effect of physical blocking by leveraging partially filled cells with the MITgcm grid (Adcroft et al., 1997). We set the 3 partial cell factors – hFacc, hFacs, hFacw – for every cell and layer based on  $\varphi$ . Due to an implementation error in the previous version of *iceberg*, the commonly enabled non-linear free surface option in MITgcm unintentionally removes the hFac values set by previous versions of *iceberg* after the initial time step, which disables the blocking effect of icebergs after the initial time step (Davison et al., 2020, 2022; Kajanto et al., 2023; Hager et al., 2024; Slater et al., 2025). We discovered and validated this effect by inspecting the output hFacc, hFacw, hFacw files at multiple time steps of model runs using the original *iceberg*. Thus, all previous studies using *iceberg* have inadvertently disabled the blocking effect after the first time step. In this updated *iceberg2* we correct this implementation error, making the package compatible with the non-linear free surface option, though the non-linear free surface solver is not utilized here, nor in previous studies.

To enable testing of the updated *iceberg2* with Hughes (2022) in Section 4, we run the model with iceberg melt disabled. This is accomplished by setting the *iceberg2* flag meltMask = False for all cells containing icebergs. In a similar fashion, for the sweep of runs considered in Section 4.2, blocking and drag are disabled together by setting the *iceberg2* flag barrierMask = False for all cells containing icebergs. This disables all blocking and drag effects from icebergs. Finally, drag alone is disabled by setting the iceberg flag barrierMask = True for all cells containing icebergs and setting the *iceberg2* variable C\_bd = 0 which makes icebergs have 0 drag, but still includes the blocking effect.

**Figure A1.** Visualization of the parameters  $\alpha, \beta$  and their variation with ice volume fraction  $\varphi$ .

We plot the variation of the available  $\alpha, \beta$  options within the new *iceberg2* package over the range of ice volume fraction  $(\varphi)$  values in Figure A1. Default values for both  $\alpha, \beta$  are indicated in the legend.

With respect to runtime improvements in this update, previously within *iceberg2* the geometry of each individual iceberg was stored as a text file and loaded for every time-step of the MITgcm. For multi-month simulations, this can result in upwards of 10<sup>6</sup> file load events. In this new iteration of *iceberg2*, at model initialization we load the geometries of all icebergs directly into memory using the READ\_REC\_3D\_RL function. While this requires slightly more memory overhead to store icebergs in memory(~10-100MB), the reduction of file load events comes with an over 90% improvement in computational performance. In a very small idealized fjord simulation (not shown), run time was reduced from 2038 seconds with the text-file method to 170 seconds while loading all iceberg geometries directly to memory at initialization.

# Appendix B: Selection of $C_{bd}$



We investigate a range of  $C_{bd}$  values to gauge what order of magnitude would be appropriate for  $C_{bd}$ , before identifying 0.002-0.003 as the range for most detailed consideration. We consider the full suite of model runs over  $\lambda, U$  as discussed in Section 4 for each value of  $C_{bd}$ . For each  $C_{bd}$  we compute the root mean squared error (RMSE) compared to results from Hughes (1973) for the upper 100 meters, as this focuses on the match for ocean velocities within the bulk of the mélange, as well as the RMSE for the upper 275 meters. We report the RMSE below each subplot for the upper 100, 275 meters. We find  $C_{bd} = 0.0025$  minimizes the RMSE<sub>100</sub> for the runs of  $\lambda = 0.02 - 0.4$ , U = 0.12 m/s (Figure B1 a-c) as well as the runs of  $\lambda = 0.2$ , U = 0.02 - 0.40 m/s (Figures B1 d-f). Figure B1 shares the same color legend as Figure 4. The RSME<sub>100</sub>, total root mean squared error in the upper 100 meters of ocean, across all  $\lambda, U$  cases considered for  $C_{bd} = 0.0025$ , is 6.049% of driving velocity.

Figure B1. Impact of varying  $C_{bd}$  on our fit to Hughes (2022). Primary effects are found in the upper 50 meters, where slower velocities (more negative residuals) result from higher  $C_{bd}$  values. We select  $C_{bd} = 0.0025$  as this minimizes the overall RMSE<sub>100</sub>.

# Appendix C: Asymmetry of Mélange Conditions for Fjord Scale Run


To highlight the asymmetry of behavior in the FJ\_NoBlockDrag case, we plot fresh water flux from the mélange only (no SGD included here) and split the north and south halves of the fjord into separate domains which we also plot in Figure C1 A. We similarly report glacier adjacent conditions split by north/south section of the domain in panel B. The FJ case shows very little north/south variation for both fresh water flux and glacier adjacent temperature. In contrast, the FJ\_NoBlockDrag case shows strong asymmetry with 60% more fresh water flux on the southern half compared to the northern half. This melt asymmetry is partially explained by the 0.5°C colder near glacier conditions on the northern half of the fjord, which is caused by recirculating cold, fresh melt water drawn back by a strong recirculating current like that shown in Figure 8.

Author contributions. PS developed the extension of *iceberg2*, and ran all models and produced all figures and lead the manuscript writing. BJ and AR supported the development of the study and contributed to the preparation of the manuscript.

Competing interests. The contact author has declared that none of the authors have any competing interests.

**Figure C1.** Impact of icebergs and drag on fjord level values. (a) Freshwater flux, including only mélange melt for the full 200 days. (b) Average temperature within 800 meters of the glacier terminus. (c) Average along fjord ocean velocity in the mid-fjord region averaged over the final 18 days. Dashed lines show the northern half of the fjord, dotted lines show the southern half of the fjord, solid lines are the average across the entire width. (d) Average salinity within 800 meters of the glacier terminus. (e) Average ocean speed within 800 meters of the glacier terminus. (f) Average glacier front melting rate. All frames share the color legend of panel (b).

Acknowledgements. All authors were supported by the GLACIOME Project, funded by the National Science Foundation (Grant No. 2025692 and 2025789). We acknowledge the computing resources that made this work possible provided by the Partnership for an Advanced Computing Environment (PACE) at Georgia Tech in Atlanta, GA with computing credits provided through startup from the University System of Georgia. We would like to thank research scientist Fang (Cherry) Liu for her assistance on challenges related to PACE and HPC. We would like to thank Benjamin J Davison and Kenneth G Hughes, both of whom generously offered their time and expertise to the development of this new parameterization. We would also like to thank Benjamin J Davison for welcoming co-development of the *iceberg* package.

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
