# Peer review of "Sub-grid Parameterization of Iceberg Drag in a Coupled Iceberg-Ocean Model"

_EGUsphere, 2025_

## Author Comment (AC1)

**Response to Reviewer 1**

*R1.1 The submitted manuscript develops a new parameterization of iceberg melt, blocking, and drag effects into the pre-existing iceberg package of the MITgcm. The representation of blocking and drag reduces the magnitude of the iceberg-induced freshwater flux and the spatial variability in velocity and temperature across a simulated representative fjord. The new developments also increase the computational efficiency of the iceberg package. The manuscript is well-written, and I recommend minor revisions.*

We thank the reviewer for their time and careful consideration of our manuscript.

*R1.2 Lines 85-95: I am not sure I agree with the description of the way the MITgcm treats the partial cell factors hFacC, hFacS, hFacW. The authors state that they are "reset" by the non-linear free surface calculation at each time step. In fact, the background h0FacC, etc., remain the same, and they are only rescaled by the stretching of the vertical coordinate. This is an intended feature, and I do not agree that it is something that should be corrected in the case of iceberg blockage. Iceberg blocking effects and the vertical stretching of the coordinate system should both be allowed the same time. In this vein, I also do not necessarily agree that the previous studies that used the r\* coordinate were deficient in that regard (I was not involved in any of those previous papers). Maybe the authors can revisit their case, and if they are still certain, they can present their argument more convincingly.*

The reviewer is precisely correct. The non-linear free surface option does not "reset" the hFactors (C,S,W) but rather allows for rescaling them via r\* prior to solving for pressure at each time step. This rescaling step respects the geometry defined by the bathymetry of the model (`h0FacC,S,W`), but this step does not respect the presence of icebergs as defined within the previous version of *iceberg*. Upon further investigation it appears the root of this issue is that `h0FacC,S,W` (as opposed to `hFacC,S,W`) are never set by *iceberg*, rather only `hFacC,S,W` are set to include icebergs. In this way, when the non-linear free surface method is enabled (but not necessarily used), each timestep re-calculates the hFactors in a way that ignores icebergs by using `h0FacC,S,W`, which never were updated to include iceberg effects. Our phrasing of this as a "reset" is misleading, and we now more properly describe this process and our confirmation of this effect in the revised manuscript.

This was confirmed by inspecting the hFactors from MITgcm, which are default exported state variables, at various time steps. When the non-linear free surface method is enabled in `CPP_OPTIONS.F` (but not necessarily used in `data`), the initial time step hFactors do include the effect of icebergs throughout the domain (hFactor $< 1$ where there are icebergs), but for timestep 2 and onward the hFactors do not include the effect of icebergs (hFactor $= 1$ everywhere in the ocean domain). In part inspired by this comment, it became clear that compatibility with the non-linear free surface option can be accomplished by setting `h0FacC,S,W` in addition to `hFacC,S,W` in the *iceberg* source code file `iceberg_init_fixed.f`, which we have now done. This change has been updated in the public code repository. This allows compatibility with enabling `NONLIN_FRSURF` in `CPP_OPTIONS.F`.

Importantly, this change does not impact our results, which we confirmed to be identical to numerical accuracy when `NONLIN_FRSURF` is enabled compared to the previous model runs of the new *iceberg* within this manuscript. This is expected, as nowhere do we actually utilize the options enabled by `NONLIN_FRSURF`, like `nonlinFreeSurf` and `select_rStar`. We leave these to their defaults following previous studies using *iceberg* (Davison et al., 2020, 2022, etc.) as well as the benchmarking studies of Hughes (2022, 2024).

To summarize, enabling but not using the option of `NONLIN_FRSURF` in `CPP_OPTIONS.F` had the unintended consequence of removing icebergs from the hFactors in previous versions of *iceberg*, after the first timestep. This resulted from only setting `hFacC,S,W` and not setting `hOFacC,S,W` in the initialization code `iceberg_init_fixed.f`. In the first version of this manuscript we blocked `NONLIN_FRSURF` to resolve this issue, but have now enabled compatibility with `NONLIN_FRSURF` via properly setting `hOFacC,S,W` in `iceberg_init_fixed.f`, which has no numerical impact on our results but increases compatibility for future uses. We have adjust the verbiage of the manuscript to clarify this.

To the reviewer's comment that the previous *iceberg* implementation using r* is not necessarily deficient: we note that the previous studies did not utilize r*, but rather only had the option to do so enabled. Davison et al. (2020, 2022), etc used the default `select_rStar = 0`, though they did enable the option in `CPP_OPTIONS.F`, but as discussed above this is the root of the issue. We do still make use of this hFactor effect, which we call "blocking" and we detail the separate influence of blocking and drag in section 4.2 and show that blocking alone produces anomalous acceleration of the ocean in cells containing icebergs.

*R1.3 Section 2: You should describe the underlying assumptions behind the representation of iceberg dynamics and thermodynamics in this package. You may even consider a brief introduction to the pre-existing package and its capability. For instance, how is "udrift" in line 65 of the manuscript defined? How is the iceberg drift estimated? More generally, please state clearly which iceberg properties and fluxes are assumed to be constant in time.*

We now include a more detailed description the previous version of *iceberg* and of $u_{drift}$, and clarify that iceberg geometries are held constant in time. Additionally, we more clearly state that heat, salt, and freshwater fluxes are solved for each timestep and thus can vary in time.

*R1.4 Lines 169-172: Could you explain more clearly why you need to set up the nonphysical temperature field? Is it not possible to achieve the same match to Hughes (2022) using the combination of 0°C temperature and 36.24 PSU salinity that you yourself mention?*

It is indeed possible to match the density alone using salinity, running the same experiment with only a salt gradient as shown in Figure R1 of this response. We show two cases producing an identical density gradient from either a linear variation in temperature (tGrad) or salt (sGrad), which results in identical resulting velocity fields. However, we still feel that reproducing the nonphysical temperature field is the most straightforward way to benchmark against the previous study so we have kept the temperature-gradient results. We have added a sentence explaining the original motivation for considering the temperature gradient case was for numerical efficiency, which can be helpful context, and have also added a sentence to clarify that a more physical salt gradient would produce the same results.

*R1.5 Figure 2 legends: "Anomonly" should read "Anomaly"*

We have corrected this typo in Figure 2.

*R1.6 Line 246: You may consider rephrasing "the sinusoidal nature of velocity" as "the sinusoidal profile of velocity."*

We have made this change for improved clarity.

*R1.7 Line 285: Point the reader back to Table 2.*

[Figure]

[Figure]

Figure R1: Comparison of the $U = 0.12, \overline{\lambda} = 0.20$ case of the FF model geometry. The temperature gradient (tGrad) case relies on using temperature variation to produce a density gradient, as discussed in the main text and Hughes (2022). Additionally we consider a salt gradient (sGrad) case where a linear salt gradient of 34 PSU at $z = 0$ m to 36.24 PSU at $z = 600$ m and temperature is constant $T = 0°C$ at all depths. Results from Hughes (2022) are plotted in a dashed gray line, and the driving velocity is a black dotted line. Residuals compared to Hughes (2022) are plotted as well.

We now point the reader back to table here. We agree this is a useful reminder to decode the numerous model configurations we consider.

**References**

Davison, B. J., Cowton, T., Sole, A., Cottier, F., and Nienow, P. (2022). Modelling the effect of submarine iceberg melting on glacier-adjacent water properties. *Cryosphere*, 16(4):1181–1196.

Davison, B. J., Cowton, T. R., Cottier, F. R., and Sole, A. J. (2020). Iceberg melting substantially modifies oceanic heat flux towards a major Greenlandic tidewater glacier. *Nature Communications*, 11(1):1–13.

Hughes, K. G. (2022). Pathways, Form Drag, and Turbulence in Simulations of an Ocean Flowing Through an Ice Mélange. *Journal of Geophysical Research: Oceans*, 127(6):1–17.

Hughes, K. G. (2024). Fjord circulation induced by melting icebergs. *Cryosphere*, 18(3):1315–1332.

---

## Author Comment (AC2)

**Response to Reviewer 2**

*R2.1 Review of Sub-grid Parameterization of Iceberg Drag in a Coupled Iceberg-Ocean Model by Summers et al.*

*Summers et al. present a significantly improved means of representing icebergs and ice melange in the MITgcm ocean model. The improvement comes in the representation of iceberg drag and in making the code much more efficient. They show that when compared with high-resolution ( 10m resolution) simulations that resolve individual icebergs, their parameterisation performs very well at resolutions ( 100s-1000s m) that are more practical for longer fjord simulations. In addition, they explore the impact of icebergs and iceberg drag on glacier-adjacent conditions.*

*I found the manuscript - and the work - to be rigorous and meticulous. In places it was a bit difficult to follow the large number of simulations, but I think this is unavoidable and in general the writing is clear, and the figures are excellent. The improvement in representation of icebergs is much needed and will be of use to other researchers simulating icebergs in fjords. The topic is certainly of relevance to the Cryosphere. In general, I would be happy to see this published with minor revisions.*

> We thank the reviewer for their summary and appreciate their time spent providing the following review.

*R2.2 Major comments*

*My main overarching thought is whether aspects of the manuscript in its present form might have been better suited to a more technical modeling journal such as Geoscientific Model Development or the Journal of Advances in Modeling Earth Systems. The main contribution of this work is an improved package/parameterisation for representing icebergs in fjord models, the paper is quite technical (often referencing model variables and settings), and the take-homes in terms of improved physical understanding are not too prominent. As someone who has worked with icebergs in MITgcm, I was able to follow and find the paper to be a very good contribution, but for someone who does not have this background the paper might be quite technical and dense. As such, I feel the manuscript could do with dialling back the technicality a little and bringing out the physical implications further.*

> We appreciate the reviewer's comments, particularly as we hope to make the manuscript approachable for the broad audience at The Cryosphere. To this end, we have moved many of the more technical aspects to Appendices to improve readability. We specifically have moved the implementation details of how hFactors are implemented, as well as the details of which *iceberg2* flags are set for each model run to an appendix. Additionally, we have moved discussion of the mechanism by which the previous version of *iceberg* had the blocking effect disabled, and our resolution of this issue, to an appendix as well. Finally, we have moved former figure C1 to the main document to help with the introduction of this model geometry.
>
> These changes have helped to dial back the technical details and make the manuscript more accessible to the more general audience of The Cryosphere.

*R2.3 Minor comments*

*L24 – this sentence didn't make sense to me – reword?*

> We have reworded this sentence.

*R2.4 Figure 1 – I think it would be helpful to note in the caption that each picture here is a column(?) of MITgcm cells, and define variables like delta x, delta y etc in the caption.*

We have clarified this in the caption, and agree it improves readability of the figure.

*R2.5 L70 – for clarity, perhaps say "reduced to a volume fraction occupied by icebergs, . . . "*

We have made this adjustment to increase clarity.

*R2.6 L88 – "hFacC, hFacS, hFacW" – I've come across these in using MITgcm but a more general reader won't be able to follow here without a bit more description.*

We have made this line a more general summary of this process, and have moved the more technical discussion to an appendix section to improve readability.

*R2.7 L95, 143, 147, 203. This may be preference, but in each of these places a figure is referred to that is much later in the paper. I find it awkward to flick forwards to find the one bit of that figure that is relevant without getting dragged into the rest of that figure, so maybe better to refer to the figures in order and remove these references?*

To improve reader flow, we have adjusted L95 to now reference the section where it is discussed, rather than the figure, as we agree this is a more appropriate reference. We have removed the reference to later figures on L143, L147 as they are not necessary. We have also removed the reference in L203 and instead point the reader back to this section for details when we later fully introduce the figure.

*R2.8 Eqs. 4&5 – might it be helpful to have a plot showing alpha and beta versus phi to visualise how they differ?*

We agree and have included this figure in an appendix.

*R2.9 L151 – "normal distribution of our iceberg drafts" – it isn't clear to me that the distribution is normal, given the power laws in Eq. 6 and L138. Is the distribution indeed normal?*

The distribution of drafts is drawn from normal distributions of $a, b$ in equation (6), but indeed this does not result in an overall normal distribution of iceberg drafts. We have changed this line to be more accurate referring to the distribution as generically "random" rather than "normal".

*R2.10 L157 – here (amongst other similar lines e.g., L286-287) is the sort of line that I feel is more suited to a journal like GMD/JAMES, because it is referring to specific modeling variable settings, which is not something I often see in Cryosphere papers. Might there be a way of avoiding these technicalities in the main text, for example by moving to an appendix or supplement?*

These lines have been moved to an "Computational Implementation" appendix to reduce the technicality of the main manuscript and increase readability for the general audience of The Cryrosphere.

*R2.11 Tables 1 & 2 – these are great and really helped me follow the results.*

We are very glad to hear these are useful to the reader. Candidly, they are very useful to us as authors as well.

*R2.12 L164 (or near here) – it would be good to refer the reader to Fig. 2A at this point.*

We have inserted a reference to Figure 2A here.

*R2.13 L169 – the non-physical temperature field – I follow what you're trying to do, but is there a reason not to use salinity to get the required stratification, which would be more physical?*

[Figure]

[Figure]

Figure R1: Comparison of a salt gradient (sGrad) and temperature gradient (tGrad) for the $U = 0.12, \overline{\lambda} = 0.20$ case. Results from Hughes (2022) are plotted in a dashed gray line, and the driving velocity is a black dotted line. Residuals compared to Hughes (2022) are plotted as well.

> This comment was also made by Reviewer 1 (R1.4), and we agree that a salinity gradient can reproduce the behavior of this non-physical temperature gradient case. We run such an experiment and show the results in Figure R1 of this reply, where we compare results for a salt gradient set-up compared to the presented temperature gradient set-up for the $U = 0.12, \overline{\lambda} = 0.20$ case. The cases produce an identical density gradient from either a linear variation in temperature (tGrad) or salt (sGrad), which results in identical resulting velocity fields. However, we find that simply reproducing the non-physical temperature field is the most straightforward way to benchmark against the previous study so we have kept the temperature-gradient results, but have added a sentence to clarify that a more realistic salt gradient would produce the same results.

*R2.14 L176 – background diffusivity seems to be set twice here, or perhaps I am misreading, but reword for clarity.*

> Indeed, the first mention of diffusivity should be viscosity. We have corrected this error.

*R2.15 L186 – this starts the description of melting before you've introduced the simulation that uses it (the melange melt second benchmark), right? If so, it would be better to move this description of melting into the paragraph below.*

> Our first model runs including melt are within this Forced Fjord domain (figure 6, experiments FF, FF_NoDrag, FF_noBlockDrag) and so we leave this description here.

*R2.16 L189 – I'm confused about this definition, or perhaps just the notation – don't lower case gammas usually denote an exchange velocity, so that there would be a velocity in this definition (see e.g., Holland & Jenkins, 1999, Journal of Physical Oceanography)?*

> This is indeed confusing terminology. Consistent with the notation of Hughes (2024), we use lower case gamma to represent the Stanton number, though we agree the exchange velocity is more commonly represented with a lower case gamma. We have clarified this with additional text to explain that this is the Stanton number, which is then multiplied by velocity to arrive at an exchange velocity, and that we are following the notation of Hughes (2024)

*R2.17 L203 – "where horizontal resolution is 10 meters" – I guess this line is a mistake since the resolution is presumably coarser and it says so on L207?*

This is indeed a typo, and we have struck this entire sentence, opting to only detail horizontal resolution in the sentence formerly on line 207.

*R2.18 L214 – In table 1 this value is C_bd = 0.025 – check for consistency through the manuscript.*

The table should read C_bd = 0.0025. We have corrected this error across the manuscript as well.

*R2.19 L222 – normalized*

adjusted

*R2.20 L241 – should this be Figure 4, not 3?*

The intent here is to guide the readers to Figure 3 D, where we show the full depth velocities to illustrate how below 300 meters the velocity field is not majorly impacted by the mélange. We have moved this reference up in the sentence and added a panel reference to make this more clear.

*R2.21 Figure 4 – I found it hard to see the line denoting the driving velocity.*

We have adjusted this line to be a bolder, black dotted line in Figures 4, 5, 6.

*R2.22 Figure 5 – it's very cool how well the package performs even at coarser resolution.*

We agree with the reviewer and were pleasantly surprised to see how well this parameterization scales at coarser resolution, even those well beyond our intended focus.

*R2.23 L326 – I feel this repeats earlier material unnecessarily.*

We have reduced and merged the text in this paragraph to avoid unnecessary repetition.

*R2.24 L330 – I didn't see the terminology "High Melt" used in section 3.1*

We now introduce this terminology in Section 3.1.

*R2.25 Fig 8b-d – at what time through the simulation are these results extracted? Since there is time variability shown in panel a, this would be relevant information to include in the text and/or caption.*

We have added this important information to the caption as well as in the main text. These are the average values of the final 2 hours of the simulation.

*R2.26 L356 – unnecessary comma and I presume "(D)." is a mistake.*

We have removed this comma and moved "(D)" up in the sentence and included the figure number to clarify that this sentencing is motivating panel D of the figure.

*R2.27 L357 – highlights*

adjusted

*R2.28 L370 (or start of this section in general) – it would be great to refer to Fig. 9 at appropriate points here.*

We now mention the Figure 10 (formerly Figure 9) in the opening of this section.

*R2.29 L421 and 489 – isn't it more than 11,000x the computational expense? If there are 40 (=400/10) more grid cells in x and y, and the timestep presumably has to be ∼ 40 times smaller, then the change in computational expense would be 40^3 = 64,000?*

The reviewer's comments is correct for linearly scaling timestep size, but the partially filled cells of *iceberg* require smaller time steps for stability than entirely full/empty cells. Practically, this means for a 400 meter resolution *iceberg* style run we require time steps of $\sim 15-25$ seconds, only $\sim 10\times$ the $1-2$ seconds in Hughes (2024). This scaling detail is not obvious but is mentioned in section 3.1. Given this is still a rough comparison, we have relaxed our stated improvement to the rounded value of $10000\times$. The value of $11000\times$ also accounted for the fact that Hughes (2024) used 50 vertical layers, rather than the 64 we consider, when requiring time steps of 2 seconds. The $11000\times$ estimate also accounted for the slightly slower per-grid-cell performance of *iceberg* ($\sim 50-150\%$ slower) which must perform melt calculations for every iceberg in the cell (up to hundreds of icebergs), rather than just 1 melt calculation per cell in the discrete iceberg approach. This estimated comparison was done by comparing the actual run times of FJ and using FJ_NoIcebergs as a lower bound of the time to run a discrete iceberg run, but perhaps this level of granularity is not important for our generalized scaling argument here.

*R2.30 L461 – minimum melt speed of 0.04 m/s*

We have corrected this typo.

*R2.31 L429 – Jain et al. preprint – it would be appropriate to cite this somewhere in the introduction.*

Jain et al. is now referenced in the introduction with the Hughes $\sim 10^1$ meter resolution models.

*R2.32 Figure C1 – since the equivalent figures for the other 2 sets of simulations are in the main paper, perhaps this could be brought into the main paper?*

We appreciate the reviewer's suggestion, and have moved this figure to the main text.

**References**

Davison, B. J., Cowton, T., Sole, A., Cottier, F., and Nienow, P. (2022). Modelling the effect of submarine iceberg melting on glacier-adjacent water properties. *Cryosphere*, 16(4):1181–1196.

Davison, B. J., Cowton, T. R., Cottier, F. R., and Sole, A. J. (2020). Iceberg melting substantially modifies oceanic heat flux towards a major Greenlandic tidewater glacier. *Nature Communications*, 11(1):1–13.

Hughes, K. G. (2022). Pathways, Form Drag, and Turbulence in Simulations of an Ocean Flowing Through an Ice Mélange. *Journal of Geophysical Research: Oceans*, 127(6):1–17.

Hughes, K. G. (2024). Fjord circulation induced by melting icebergs. *Cryosphere*, 18(3):1315–1332.